

# Moderation effects of food intake on the relationship between urinary microbiota and urinary interleukin-8 in female type 2 diabetic patients

Fengping Liu[1,*], Zongxin Ling[2,*], Chulei Tang[3], Fendi Yi[4] and Yong Q. Chen[1]

[1] Wuxi School of Medicine, Jiangnan University, Wuxi, Jiangsu, China
[2] Collaborative Innovation Center for Diagnosis and Treatment of Infectious Diseases, State Key Laboratory for Diagnosis and Treatment of Infectious Diseases, The First Affiliated Hospital, School of Medicine, Zhejiang University, Hangzhou, Zhejiang, China
[3] Xiangya Nursing School, Central South University, Changsha, China
[4] Endocrinology Department, The Affiliated Yancheng Hospital of Southeast University Medical College, Yancheng, China
* These authors contributed equally to this work.

Corresponding authors
Fendi Yi, wygfsjk123@yeah.net
Yong Q. Chen,
yqchen@jiangnan.edu.cn

## ABSTRACT

**Background:** Our previous study demonstrated that the composition of the urinary microbiota in female patients with type 2 diabetes mellitus (T2DM) was correlated with the concentration of urinary interleukin (IL)-8. As the composition of urine is mainly determined by diet, diet might mediate the correlation.

**Methods:** Seventy female T2DM patients and 70 healthy controls (HCs) were recruited. Midstream urine was used for the urine specimens. Urinary IL-8 was determined by enzyme-linked immunosorbent assay. A Chinese Food Frequency Questionnaire was used to collect food intake data. The independent variables in the hierarchical regression analysis were the relative abundances of the bacterial genera and species that were significantly different between the T2DM and HCs and between the T2DM patients with and without detectable urinary IL-8, and the bacterial genera associated with IL-8 concentration in the multiple regression model reported in our previous research. IL-8 concentration was the dependent variable, and nutrient intakes were moderator variables.

**Results:** Fiber and vitamin B3 and E intake exerted enhancing effects, and water intake exerted a buffering effect, on the positive relationship between the relative abundance of *Ruminococcus* and IL-8 concentration ($p < 0.05$). Cholesterol and magnesium intake exerted enhancing effects on the positive relationship between the relative abundance of *Comamonas* and IL-8 concentration ($p < 0.05$).

**Conclusion:** Modulating T2DM patients' dietary patterns may prevent bladder inflammation.

## INTRODUCTION

Diabetes mellitus (DM) is a severe chronic disease that is, recognized as a global public health problem. In recent decades, the prevalence of DM has increased substantially. Globally, about 1 in 11 adults have DM (90% have type 2 diabetes mellitus (T2DM)) (*Zheng, Ley & Hu, 2018*). Infections were more common prevalent with T2DM than among respondents without diabetes (*Carey et al., 2018*), especially urinary tract infection (UTI) (*Wilke et al., 2015*). UTI is of major concern, and many studies have demonstrated an increased prevalence of UTI in T2DM patients (*Wilke et al., 2015*; *Hirji et al., 2012*; *Nichols et al., 2017*), especially in female patients (*Hirji et al., 2012*). Traditionally, UTI has been thought to be caused by bacterial invasion of the patient's bladder. However, studies on the urinary microbiota have caused this concept to be questioned over the past decade, as DNA sequencing and expanded quantitative urine culture technique have shown that bacteria live in almost all individuals' bladders, both UTI patients and healthy subjects (*Price et al., 2016*; *Thomas-White et al., 2018*). Therefore, the presence of bacteria in the bladder of UTI patients may just indicate an altered bacterial profile compared to that of healthy individuals rather than the invasion of new bacteria.

An increasing body of evidence suggests that T2DM is associated with profound gut dysbiosis (*Tai, Wong & Wen, 2015*; *Wang et al., 2017*). Food intake patterns can modulate the dysbiosis, which in turn impacts metabolic health (*David et al., 2014*; *Simpson & Campbell, 2015*). The kidneys are primary excretory organs, and are responsible for the elimination of waste metabolites. Food intake patterns determine the metabolites in urine, which may affect the bacterial profile in the bladder (*Shields-Cutler et al., 2015*).

It has been reported that interleukin (IL)-8 can function as a diagnostic biomarker for UTI, with a sensitivity of 93% and a specificity of 90%, which is much greater than the sensitivity and specificity of urine culturing (*Rao, Evans & Finn, 2001*). Similarly, another study demonstrated that urinary IL-8 could be considered as a surrogate marker for the rapid diagnosis of bacteriuria (*Zaki, 2008*). Therefore, we detected the urinary IL-8 concentration in our previous research, and found that 46 out of 70 subjects had detectable concentrations of urinary IL-8 (*Ling et al., 2017*).

Our previous research demonstrated that T2DM patients had lower bacterial diversity in urinary microbiome than healthy controls (HCs). There were significant differences in the relative abundance of multiple urinary bacterial genera between T2DM patients and HCs, including *Lactobacillus*, *Prevotella* and *Pseudomonas* (*Liu et al., 2017*). Subgroup analysis showed that T2DM patients with detectable concentrations of urinary IL-8 (WIL8) had a different urinary bacterial profile from those with no detectable concentrations of urinary IL-8 (NIL8); indeed, the WIL8 group had notable elevations and reductions in several bacterial genera and species compared to the NIL8 group. Moreover, subjects in the T2DM group with higher or lower relative abundance of specific urinary bacteria genera, relative to HCs, had different concentrations of IL-8. Additionally, several urinary genera were significantly associated with the urinary IL-8 concentration in T2DM patients, including *Ruminococcus*, *Anaerotruncus* and *Lactobacillus*. Moreover,

the WIL8 group had significantly higher urinary concentrations of nitrites and leukocyte esterase than the NIL8 group (*Ling et al., 2017*).

As stated above, the urinary microbiome profile in T2DM females is different from HCs, and the bacterial dysbiois is related to the expression level of IL-8. Studies on gut microbiome have shown that diet regulate the composition of microbiome and inflammation (*Tilg & Moschen, 2015*). On one hand, gut microbiome can rapidly respond to altered diet. *Walker et al. (2011)* demonstrated that increased intake of resistant starch can substantially alter the species composition of the colonic microbiome. *David et al. (2014)* reported that short-term consumption of diets composed entirely of animal or plant products resulted in microbial community structure. On the other hand, studies have shown that intestinal dysbiosis led to an abnormal adaptive immune response that increased inflammatory gut diseases (*Seksik et al., 2003*). However, to date, no studies have shown whether the relationship between urinary microbiome and IL-8 level is regulated by nutrients intake. Therefore, our present study aimed to analyze the regulating role of nutrient intake on the correlation between the urinary microbiome and IL-8 using moderation effect model. If nutrients intake has a moderating role, clinicians can precisely adjust patient's dietary composition based on the correlation between urinary microbiome and IL-8 concentration.

## MATERIALS AND METHODS

### Recruitment of subjects

Briefly, a case-control design was used in our previous study, in which female T2DM patients were individually matched to HCs according to age, marital status, and menstrual status. The same data were used in the present study. The participants were recruited from the First Affiliated Hospital, School of Medicine, Zhejiang University, from 2 July 2015 to 2 January 2016. For every 10 T2DM patients recruited, 10 HCs who matched the patients' age, marriage and menstrual status were recruited. Subjects with the following attributes were excluded: UTI in the previous month; use of antibiotics, probiotics, prebiotics, or synbiotics in the previous 3 months; inability to complete the questionnaire; menstruation; urinary incontinence; known anatomic urinary tract abnormalities (e.g., cystoceles, hydronephrosis, renal atrophy, or neurogenic bladder); or urinary catheter use. The Ethics Committee of the First Affiliated Hospital, School of Medicine, Zhejiang University, approved the study (Ref. 295). Written informed consent was obtained from each patient before enrollment.

### Urine specimen collection, DNA isolation and sequencing and IL-8 detection

A self-designed midstream urine collection technique, which was reported in our previous study, was used to collect the first urine of the day (*Liu et al., 2017*). Bacterial DNA was isolated using magnetic beads, and the 16S rRNA gene V3–V4 regions of microbial genomic DNA were amplified by PCR (forward primer, 5′-ACTCCTACGGGAGGCAG CAG-3′; reverse primer, 5′-GGACTACHVGGGTWTCTAAT-3′). Amplicons were then normalized, pooled, and sequenced using an Illumina MiSeq desktop sequencer

(2 × 300 bp paired-end run). Sequence read processing was performed using QIIME (version 1.9.0), which included quality trimming and demultiplexing. Operational taxonomic unit (OTU) selection, using Usearch (version 1.11.1), included dereplication (—derep_fullength), clustering (—cluster_fast, —id 0.97), and chimera detection (—uchime_ref). Taxonomic assignment for individual datasets was conducted using Greengene (version 13.8). The output file was further analyzed using Statistical Analysis of Metagenomic Profiles software (version 2.1.3). Urinary IL-8 concentrations were determined using enzyme-linked immunosorbent assay kits (RayBiotech, Inc., Norcross, GA, USA).

## Food intake assessment and nutrient intake calculations

The influence of diet on the human microbiota is long term, so almost all previous studies on the relationship between the human microbiota and nutrition used the Food Frequency Questionnaire (FFQ) (*Mandal et al., 2016*; *Ribeiro et al., 2017*; *Singh et al., 2017*), which requires participants to recall food intake in the previous year. Therefore, a Chinese version of the FFQ was used to assess food intake (*Zhao et al., 2010*). Prior to the present study, we conducted a pre-experimental study. We evaluated the reproducibility of the FFQ (*Chavarro et al., 2009*), in which 10 volunteers were invited to filled the FFQ for twice, for example, $FFQ_1$ and $FFQ_2$. The interval time between the two interviews was 2 weeks $FFQ_1$ and $FFQ_2$. Then we assessed the correlation coefficient of nutrients intake of energy, protein, fat and carbohydrate, and found they were significantly correlated in $FFQ_1$ and $FFQ_2$ ($p < 0.05$; data was not shown) indicating that the reproducibility was good. In the meantime, we conducted a qualitative interview. Based on their suggestions, a face-to-face interview was carried out in each participant's kitchen, where the sizes of the participant's dinnerware items were evaluated. The participants were asked to recall their food and water intake patterns over the previous year (up to the day of urine sample collection). Before the interviews, we visited local supermarkets to convert the quantities of certain foods to weights, as most participants were able to remember the amount, but not the weight, of the food they ate. Subsequently, we separated the elements of the reported food intake into large, moderate, and small portion sizes. Data on the intake of Chinese herbs and supplements were also collected. Water intake included drinking water and the water content of foods. The mean duration for assessing each patient's food intake data was 90 min. MATLAB software (version 7.0) was used to convert the amount of food intake into daily nutrient intake.

## Statistical analysis

Statistical analysis was performed using SPSS software (version 21.0). For continuous variables, independent *t*-tests were applied. For categorical variables, either Pearson's chi-square tests or Fisher's exact tests were used depending on assumption validity. A moderation effect can be enhancing, buffering or antagonistic (*Fairchild & MacKinnon, 2009*). Therefore, hierarchical regression analysis was used to analyze the moderating effects of nutrient intake on the relationships between the urinary microbiota and IL-8 concentration. To establish hierarchical regression models that would be likely to be true

representations of real relationships, the independent variables selected were the bacterial genera and species that exhibited differences in relative abundances between the T2DM and HC groups and between the WIL8 and NIL8 groups of T2DM patients, and also the bacterial genera that contributed to the urinary IL-8 concentration in the T2DM cohort (according to our previous research). The moderator variables were nutrient intakes; the dependent variable was urinary IL-8 concentration; and the control variables were fasting blood glucose, urine glucose, age, menstrual status, and body mass index (*Liu et al., 2017*; *Ling et al., 2017*). Before conducting hierarchical regression analysis, Pearson's correlation analysis was used to determine whether the independent and dependent variables were significantly correlated. The independent variables that were significantly correlated with the dependent variable were then entered into hierarchical regression models. The hierarchical regression analysis featured three-steps. First, we entered age, body mass index, fasting blood glucose, urine glucose, and menstrual status as control variables, as our previous study showed that these factors affected the urinary microbiota. Second, we entered other independent variables, such as the relative abundances of bacteria and nutrient intakes. Third, we entered the "bacteria × nutrient" interaction terms. The regression effects were then analyzed. All tests of significance were two-sided, and $p < 0.05$ was considered significant. After obtaining β values for the control variables, main effects, and interaction effects, ModGraph-1 software was used to calculate cell means to display moderation effects in a graphical manner. ModGraph-1 software calculated the effects of high, medium, and low nutrient intake on the relationships between the relative abundance of bacteria and IL-8 concentration.

## RESULTS

### Patient characteristics, DNA sequencing and IL-8

Seventy T2DM patients were recruited. Their characteristics are shown in Table 1. A total of 3,981,519 reads were obtained, accounting for 76.93% of the valid reads. The mean read length was 438 bp (range, 423–486 bp). Good's coverage estimator was 98.00%. Sequence data from this study are deposited in the GenBank Sequence Read Archive (Accession No. SRP087709). The mean urinary IL-8 concentration was 42.26 ± 64.66 pg/mL (*Liu et al., 2017*; *Ling et al., 2017*).

### Several nutrients exhibited moderation effects on the associations between the urinary microbiota and urinary IL-8 concentrations

In our previous studies, the relative abundance of 33 bacterial genera-level and 9 bacterial species-level OTUs showed significant differences between T2DM patients and HCs (Figs. S1 and S2), 21 bacterial genera-level and 10 bacterial species-level OTUs showed significant differences between the WIL8 and NIL8 T2DM subgroups (Figs. S3 and S4), and 18 bacterial genera were predictors of urinary IL-8 in a stepwise regression analysis (Table S1). In addition, we identified T2DM patients with significant differences in their urinary microbiota compared with HCs, grouping them according to the bacterial genera present in their urine samples ("≥HCs" or "<HCs"). We then analyzed the urinary IL-8 concentrations for the T2DM patients with significantly high or low abundance of a

**Table 1  Characteristics of T2DM subjects.**

| Parameters | Value for corhort ($n^a$)$^b$ or statistic | | $p$ value$^c$ |
|---|---|---|---|
| | T2DM ($n$ = 70) | HC ($n$ = 70) | |
| Age (yr) | | | 1.00 |
| 26–35 | 2 (2.86) | 2 (2.86) | |
| 36–45 | 6 (8.57) | 6 (8.57) | |
| 46–55 | 11 (15.71) | 11 (15.71) | |
| 56–65 | 17 (24.3) | 17 (24.3) | |
| 66–75 | 23 (32.9) | 23 (32.9) | |
| ≥70 | 11 (15.7) | 11 (15.7) | |
| Marital status (no. (%)) | | | 1.00 |
| Living with couple | 64 (91.4) | 64 (91.4) | |
| Living without couple | 6 (8.6) | 6 (8.6) | |
| Menstrual status (no. (%)) | | | 1.00 |
| Premenopause | 11 (15.7) | 11 (15.7) | |
| Postmenopause | 54 (77.1) | 54 (77.1) | |
| Hysterectomy | 5 (7.1) | 5 (7.1) | |
| BMI (kg/m$^2$) | 23.87 ± 3.65 | 23.10 ± 4.49 | 0.27 |
| Exercise$^d$ | | | 0.15 |
| Class 1 | 3 (4.29) | 4 (5.71) | |
| Class 2 | 19 (27.14) | 17 (24.29) | |
| Class 3 | 19 (27.14) | 30 (42.86) | |
| Class 4 | 26 (37.14) | 14 (2.00) | |
| Class 5 | 3 (4.29) | 5 (7.14) | |
| Duration of T2DM (yr) | 9.77 ± 7.49 | NA | |
| FBG (mmol/L) | 7.83 ± 2.35 | 5.22 ± 0.61 | 0.00 |
| Urine pH | 5.81 ± 0.62 | 5.93 ± 0.70 | 0.28 |
| Urine protein POS (no. (%)) | 14 (20.00) | 3 (4.00) | 0.01 |
| Urine nitrite POS (no. (%)) | 6 (8.57) | 4 (5.71) | 0.75 |
| Urine leukocyte esterase POS (no. (%)) | 13 (18.57) | 18 (25.71) | 0.42 |
| Urine glucose POS (no. (%)) | 14 (20.00) | 0 (0.00) | 0.00 |
| UTI in the past 1 year (no. (%)) | 27 (38.57) | 7 (10.00) | 0.00 |
| Urine culture (*E. coli*) POS (no. (%)) | 6 (8.57) | 5 (7.14) | 1.00 |

Notes:
[a] n, no. of subjects.
[b] Mean ± SD or no. (%).
[c] Pearson's chi-square test or Fisher's exact test was used for categorical variables and $t$ test was used to compare continuous variables.
[d] Taking exercise: Class 1: never; Class 2: several times per year; Class 3: several times per month, but every week; Class 4: several times per week, but every day; Class 5: every day.
BMI, body mass index; *E. coli*, *Escherichia coli*; FBG, fasting blood glucose; HC, healthy control; NEG, negative; POS, positive; T2DM, type 2 diabetes mellitus; UTI, urinary tract infections; yr, year.

given genus, relative to the HCs. The IL-8 concentration was low in T2DM patients with high *Acinetobacter*, *Microbacterium* and *Megamonas* abundance, but high in T2DM patients with high *Pseudomonas* and *Klebsiella* abundance (Fig. S5) (*Liu et al., 2017*; *Ling et al., 2017*). These bacterial genera were included in the hierarchical regression analysis (Table S2).

**Table 2  Comparison of nutrient intake between T2DM and HCs.**

| Nutrients | T2DM | HCs | *p* value |
|---|---|---|---|
| Water (mL/d) | 2,526.04 ± 903.79 | 2,484.24 ± 829.94 | 0.79 |
| Energy (kcal/d) | 1,256.55 ± 479.72 | 1,469.07 ± 626.63 | 0.03 |
| Protein (g/d) | 60.73 ± 27.29 | 72.85 ± 46.09 | 0.06 |
| Fat (g/d) | 57.75 ± 30.05 | 59.32 ± 29.40 | 0.76 |
| SFA (% TE/d) | 11.65 ± 6.39 | 13.73 ± 11.29 | 0.18 |
| MUFA (% TE/d) | 18.67 ± 11.96 | 19.15 ± 11.32 | 0.81 |
| PUFA (% TE/d) | 16.71 ± 10.68 | 15.10 ± 8.68 | 0.33 |
| Carbohydrate (g/d) | 151.46 ± 58.59 | 184.38 ± 94.23 | 0.01 |
| Fiber (g/d) | 9.26 ± 4.54 | 12.50 ± 17.20 | 0.13 |
| Cholesterol (mg/d) | 472.36 ± 212.18 | 486.47 ± 179.68 | 0.67 |
| Vitamin A (μgRE/d) | 745.95 ± 315.96 | 1,032.52 ± 1,351.94 | 0.09 |
| Retinol (μg/d) | 265.07 ± 178.10 | 380.30 ± 632.97 | 0.15 |
| Vitamin B1 (mg/d) | 0.70 ± 0.32 | 0.77 ± 0.32 | 0.18 |
| Vitamin B2 (mg/d) | 0.88 ± 0.33 | 1.18 ± 0.93 | 0.01 |
| Vitamin B3 (mg/d) | 13.92 ± 6.84 | 18.77 ± 15.09 | 0.02 |
| Vitamin C (mg/d) | 107.44 ± 40.45 | 127.10 ± 62.21 | 0.03 |
| Vitamin E (mg/d) | 13.83 ± 42.49 | 7.61 ± 20.65 | 0.27 |
| Calcium (mg/d) | 687.33 ± 323.02 | 831.39 ± 681.37 | 0.11 |
| Phosphorus (mg/d) | 936.87 ± 388.41 | 1,090.99 ± 547.12 | 0.06 |
| Potassium (mg/d) | 1,704.71 ± 678.50 | 2,262.05 ± 1,882.30 | 0.02 |
| Sodium (mg/d) | 2,859.29 ± 1586.14 | 2,788.47 ± 1,585.96 | 0.79 |
| Magnesium (mg/d) | 267.73 ± 116.61 | 325.39 ± 233.91 | 0.07 |
| Iron (mg/d) | 21.47 ± 10.75 | 26.91 ± 24.19 | 0.09 |
| Zinc (mg/d) | 9.49 ± 3.84 | 11.47 ± 6.54 | 0.03 |
| Selenium (μg/d) | 38.42 ± 17.68 | 49.71 ± 29.51 | 0.01 |
| Copper (mg/d) | 1.78 ± 1.00 | 2.32 ± 2.47 | 0.10 |
| Manganese (mg/d) | 4.17 ± 1.80 | 8.95 ± 32.13 | 0.23 |

**Note:**
TE, total energy intake; RE, retinol equivalent; SFA, saturated fatty acid; MUFA, monounsaturated fatty acid; PUFA, polyunsaturated fatty acid.

The relative abundances of *Ruminococcus*, *Comamonas*, *Cytophaga*, *Providencia*, *Anaerotruncus*, *Giesbergeria*, *Meiothermus* and *Luteibacter* were correlated with increased urinary IL-8 concentrations, whereas *Cloacibacterium* was correlated with decreased IL-8 (Table S3). When these bacteria were used as independent variables, urinary IL-8 was used as the dependent variable, and nutrient intake variables were used as moderating variables (Table 2), hierarchical regression analysis showed that several nutrients moderated the correlation between the urinary microbiota and IL-8.

The relative abundance of *Ruminococcus* and water intake had significant effects on urinary IL-8 concentration (Table S4). The "*Ruminococcus* × Water" interaction effect was also significant, and it explained 39.00% of the variance in IL-8 concentration. The strongest negative association between the relative abundance of *Ruminococcus* and IL-8 concentration occurred in patients who reported a high water intake, while the

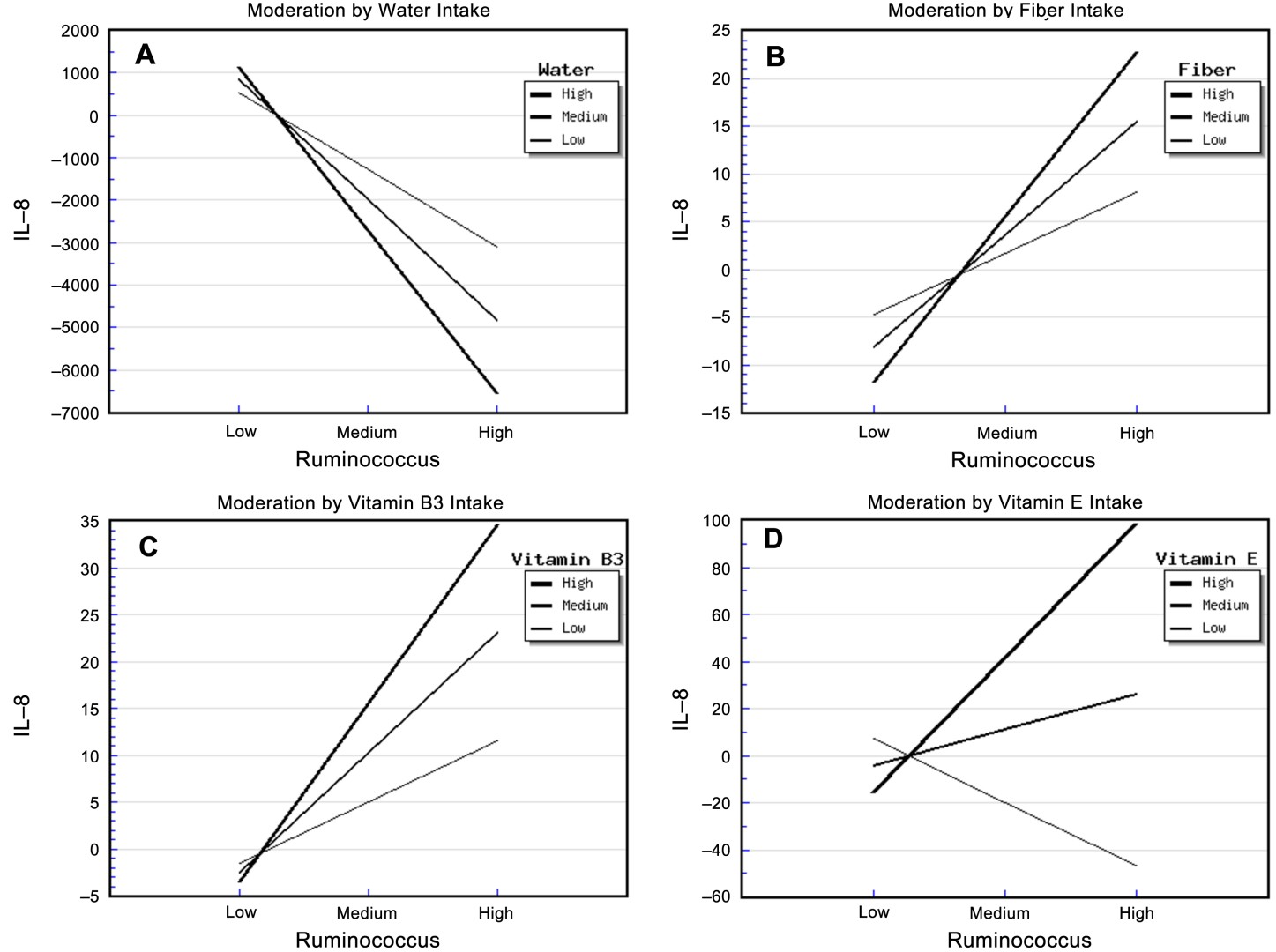

**Figure 1 Diet, *Ruminococcus* and IL-8.** The steepest slope represents the strongest moderation effect, while the flattest slope represents the weakest moderation effect. The downward lines indicate negative moderation effects. (A) Moderation by water intake. The high water intake value was 2526.0 + 903.79 mL/d, the medium value was 2526.04 mL/d, and the low value was 2526.04 − 903.79 mL/d. (B) Moderation by fiber intake. The high fiber intake value was 9.26 + 4.54 g/d, the medium value was 9.26 g/d, and the low value was 9.26 − 4.54 g/d; (C) Moderation by vitamin B3 intake. The high vitamin B3 intake value was 13.92 + 6.84 mg/d, the medium value was 13.92 mg/d, and the low value was 13.92 − 6.84 mg/d. (D) Moderation by vitamin E intake. The high vitamin E intake value was 13.83 + 42.49 mg/d, the medium value was 13.83 mg/d, and the low value was 13.83 − 42.49 mg/d.

weakest negative association occurred for patients who reported a low water intake (Fig. 1A). This suggested that water intake buffered the relationship between the relative abundance of *Ruminococcus* and IL-8 concentration.

The relative abundance of *Ruminococcus* and fiber intake had significant effects on urinary IL-8 concentration (Table S5). The "*Ruminococcus* × Fiber" interaction effect was also significant, and it explained 41.70% of the variance in IL-8 concentration. In addition, the strongest positive association between the relative abundance of *Ruminococcus* and IL-8 concentration occurred in patients who reported a high fiber intake, whereas the weakest association occurred in those who reported a low fiber intake (Fig. 1B). This

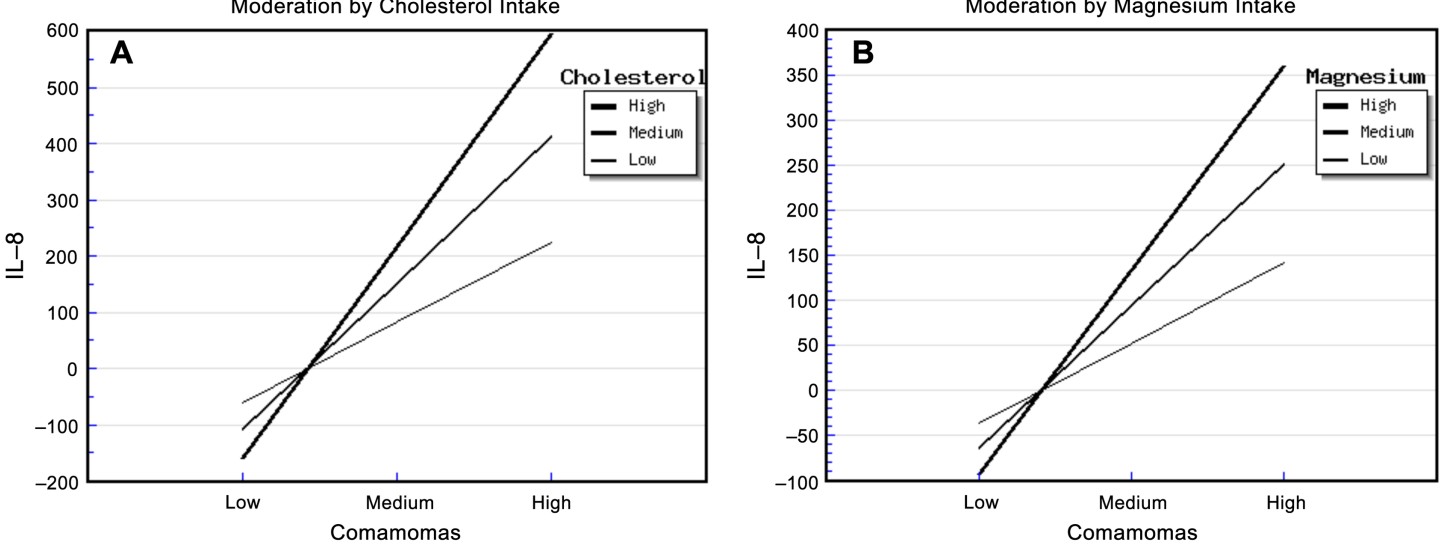

**Figure 2  Diet, *Comamonas* and IL-8.** The steepest slope represents the strongest moderation effect, while the flattest slope represents the weakest moderation effect. (A) Moderation by cholesterol intake. The high cholesterol intake value was $472.36 + 212.18$ mg/d, the medium value was $472.36$ mg/d, and the low value was $472.36 - 212.18$ mg/d. (B) Moderation by magnesium intake. The high magnesium intake value was $267.73 + 116.61$ mg/d, the medium value was $267.73$ mg/d, and the low value was $267.73 - 116.61$ mg/d.

indicated that fiber intake enhanced the positive correlation between the relative abundance of *Ruminococcus* and IL-8 concentration.

In addition, the relative abundance of *Ruminococcus* and vitamin B3 intake had significant effects on urinary IL-8. The "*Ruminococcus* × Vitamin B3" interaction effect was also significant, and it explained 32.00% of the variance in IL-8 concentration (Table S6). The strongest positive association between the relative abundance of *Ruminococcus* and IL-8 concentration occurred in patients who reported a high vitamin B3 intake, whereas the weakest association occurred in patients who reported a low vitamin B3 intake. This indicated that vitamin B3 intake enhanced the positive correlation between the relative abundance of *Ruminococcus* and IL-8 concentration (Fig. 1C).

The relative abundance of *Ruminococcus* and vitamin E intake also had significant effects on urinary IL-8. The "*Ruminococcus* × Vitamin E" interaction effect was also significant, and it explained 34.40% of the variance in IL-8 concentration (Table S7). Surprisingly, the strongest positive association between the relative abundance of *Ruminococcus* and IL-8 concentration occurred in patients who reported a high vitamin E intake, the weakest association occurred in patients who reported a medium vitamin E intake, and a negative association was apparent in patients who reported a low vitamin E intake (Fig. 1D). This suggests that not only did high vitamin E intake enhance the relationship between the relative abundance of *Ruminococcus* and IL-8 concentration, but also that the amount of vitamin E intake moderated this relationship in a highly sensitive manner.

Vitamin C intake had a significant main effect in the regression model involving the relationship between the relative abundance of *Ruminococcus* and IL-8 concentrations, but there was no "*Ruminococcus* × Vitamin C" interaction effect (Table S8).

The relative abundance of *Comamonas* and cholesterol intake had significant effects on urinary IL-8. The "*Comamonas* × Cholesterol" interaction effect was also significant, and it explained 68.00% of the variance in IL-8 concentration (Table S9). The strongest positive association between the relative abundance of *Comamonas* and IL-8 concentration occurred in patients who reported a high cholesterol intake, and the weakest association occurred in patients who reported a low cholesterol intake (Fig. 2A). This suggested that cholesterol intake enhanced the relationship between the relative abundance of *Comamonas* and IL-8.

Similarly, the relative abundance of *Comamonas* and magnesium intake had significant effects on urinary IL-8. The "*Comamonas* × Magnesium" interaction effect was also significant, and it explained 73.00% of the variance in IL-8 concentration (Table S10). The strongest positive association between the relative abundance of *Comamonas* and IL-8 concentration occurred in patients who reported a high magnesium intake, and the weakest association occurred in patients who reported a low magnesium intake (Fig. 2B). This suggested that magnesium intake enhanced the relationship between *Comamonas* and IL-8 concentration.

## DISCUSSION

During the last decade, evidence has accumulated to support a role for the microbiota, including the bacterial communities in the gut and oral cavity, in T2DM patients (*Navab-Moghadam et al., 2017*; *Larsen et al., 2010*; *Zhang et al., 2018*). Not only do T2DM patients have a different microbiota profile from healthy subjects, but alterations in the microbiota are also responsible for immune responses, such as the expression of proinflammatory cytokines (*Xiao et al., 2017*; *Pushpanathan et al., 2016*). Both human and animal studies have shown that diet contributes to the composition of the gut microbiota (*Egshatyan et al., 2014*; *Wen & Duffy, 2017*), and a gut microbiota signature that promotes intestinal inflammation may exist, thereby promoting the development of T2DM (*Wen & Duffy, 2017*). Similar to previous findings relating to the gut and oral microbiota, our previous research demonstrated that T2DM patients have a different urinary microbiota composition from HCs, and the urinary microbiota composition was correlated with urinary IL-8 concentration in T2DM patients (*Liu et al., 2017*; *Ling et al., 2017*). In accordance with the "diet—gut microbiota—proinflammatory response" pathway described in previous work, we investigated whether diet moderates the relationship between the urinary microbiota and IL-8 concentration.

In this study, the relative abundance of *Ruminococcus* was positively associated with urinary IL-8. *Ruminococcus* spp. have rarely been reported in human urinary microbiota studies. A previous human urinary microbiome study demonstrated that *Ruminococcus* was positively associated with preterm birth (*Ollberding et al., 2016*), and it was detected in the urine of patients with chronic allograft dysfunction (*Wu et al., 2018*). These indicate that urinary *Ruminococcus* might be associated with the occurrence and development of some diseases. In our study, none of the subjects were diagnosed with UTI at recruitment when their urine specimens were collected, but most samples having detectable *Ruminococcus* and IL-8. This together with the positive correlation

of *Ruminococcus* and urinary IL-8 level indicates that the potential of members of *Ruminococcus* as a contributor of IL-8 and UTI in T2DM patients. Future studies should investigate whether the UTI occurrence in the patients with detectable IL-8 is higher than those with non-detectalbe IL-8.

Interestingly, water intake weakened the positive relationship between the relative abundance of *Ruminococcus* and IL-8 concentration. Resolution of a UTI relies on the concentration of an antibiotic within the urine, but highly concentrated urine reduced bactericidal activity against some pathogens (*Mikel & Marta, 2003*). In a recent randomely controlled trial, increased water intake was effective on controlling the number of cystitis episodes in women with reccurrent cystitis (*Hooton et al., 2018*). Meanwhile, increased water intake can reduce the dose of antimicrobial-regimens (*Hooton et al., 2018*).

Surprisingly, increasing fiber intake enhanced the relationship between the relative abundance of *Ruminococcus* and IL-8 concentration. In the gut, increased fiber intake improves the multiplication of *Ruminococcus* (*Abell et al., 2008*), and dietary fiber was inversely associated with inflammatory markers, including C-reactive protein (*Wannamethee et al., 2009*), IL-6 (*Wannamethee et al., 2009*; *Qi et al., 2006*), TNF-α-R2 (*Qi et al., 2006*; *Ma et al., 2008*). Therefore, it seems that fiber intake contribute inversely to the relationship of *Ruminococcus* with inflammatory cytokines from the gut to the urine.

Both vitamin B3 and vitamin E intake exerted an enhancing effect on the positive relationship between *Ruminococcus* and IL-8 concentration. Since no previous studies reported that the roles of vitamin B3 and vitamine E on either *Ruminoccocus* growth and IL-8 expression, future studies should be focused on the role of the two vitamins in human urinary microbiome, then explore their moderation effects on the correlation between urinary microbiome and inflammatory markers.

Cholesterol intake had significant enhancing effects on the abundance of *Comamonas* and urinary IL-8. *Comamonas* is a bacterial genus in the phylum *Proteobacteria*. The current literatures supported that *Comamonas* spp. was related to infections including phlegmon, peritonitis, endocarditis, meningitis, septicemia and UTI (*Cooper et al., 2005*; *Biswas, Fitchett & O'Hara, 2014*; *Horowitz et al., 1990*). *Barua et al. (2017)* found that nephritic syndrome patients with higher serum cholesterol were more susceptible to UTI than those with normal serum cholesterol, which might be because high cholesterol concentrations suppress lymphocyte function. With regard to magnesium, its intake had an enhancing effect on the relationship between the relative abundance of *Comamonas* and IL-8 concentration. However, it is not recommended that T2DM patients should restrict magnesium intake, since a previous in vitro study demonstrated that magnesium sulfate suppressed inflammatory response by human unbilical vein cells (*Rochelson et al., 2007*; *Weglicki et al., 1992*), and a previous rodent model study demonstrated that magenesium-deficiency elevated circulating levels of inflammatory cytokines (*Weglicki et al., 1992*).

There were several limitations in our present study. Firstly, the participants were only from Zhejiang Province, China. It is known that different areas have different dietary patterns in China. For example, in Chinese southern areas (incuding Zhejiang province),

the body sizes of the local population are generally small and short, and their diet is light and total intake is fewer than the northern areas. Thus, a multi-centers study, including southern and northern areas in China, is needed to verify the results in the present study. Secondly, we have failed to detect the co-effect of bacteria on the correlation between urinary microbiome and inflammatory response, since the 16S rDNA can only detect at the bacterial genus level and it is impossible to culture bacterial genus to examine their co-effect. Thirdly, T2DM patients might assess their dietary intake and exercise more accurately than the healthy subjects, because there were monthly healthcare education programs for patients in the communities in Zhejiang province. During the programs, the patients were instructed that dietary control and proper exercise could alleviate the progress of diabetes. Therefore, the patients might underestimate their dietary intake and overestimate their exercise. This might be one of the reasons why the patients in the present study had lower energy intake and equal exercise time than the healthy subjects, while their values of BMI were not different. Also, the response of assessment of dietary intake and and exercise might be different between the patients and healthy subjects, which might be a bias for assessing urinary microbiome and IL-8 production.

## CONCLUSIONS

To our knowledge, this study is the first study to demonstrate the moderating effects of food intake on the relationship between the midstream urinary microbiota and urinary IL-8 concentration in humans. It is suggested that modulating the dietary patterns of patients based on urinary microbiota dysbiosis status and urinary IL-8 concentration might play a role in regulating inflammation response in urinary tract system.

## ACKNOWLEDGEMENTS

We gratefully acknowledge the volunteers who participated in our study. We also thank the Charlesworth Group for editing the English in the manuscript.

### Funding

The authors received no funding for this work.

### Competing Interests

The authors declare that they have no competing interests.

### Author Contributions

- Fengping Liu conceived and designed the experiments, performed the experiments, analyzed the data, prepared figures and/or tables, authored or reviewed drafts of the paper, and approved the final draft.
- Zongxin Ling performed the experiments, analyzed the data, prepared figures and/or tables, and approved the final draft.
- Chulei Tang performed the experiments, prepared figures and/or tables, and approved the final draft.

- Fendi Yi conceived and designed the experiments, prepared figures and/or tables, and approved the final draft.
- Yong Q. Chen conceived and designed the experiments, authored or reviewed drafts of the paper, and approved the final draft.

## Human Ethics

The following information was supplied relating to ethical approvals (i.e., approving body and any reference numbers):

The study was approved by the Ethics Committee of the First Affiliated Hospital, School of Medicine, Zhejiang University (295).

## Data Availability

Data is available at NCBI: SRP087709 and PRJNA329447.

## Supplemental Information

Supplemental information for this article can be found online at http://dx.doi.org/10.7717/peerj.8481#supplemental-information.

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
