# Peer review of "Moderation effects of food intake on the relationship between urinary microbiota and urinary interleukin-8 in female type 2 diabetic patients"

_PeerJ, doi:10.7717/peerj.8481_

## Round 0.1 · original submission · Major Revisions

Reviewer 1 in particular has raised a number of substantive issues relating to the characteristics of the control groups used, and the tendency in places to infer conclusions that are not supported entirely by the data. I would urge you to carefully address these points, together with the relatively more minor issues highlighted by reviewers 2 and 3, in your revised manuscript.

·

Basic reporting

The manuscript is generally well written and conforms to the norm expected for a peer-review journal article.

Minor comments to improve:
First paragraph of introduction - DM; are the authors referring to T1DM or T2DM in the opening paragraph? This distinction should be clear.

Lines 77-83 - it would be helpful to clarify that this data being referred to is urinary microbiome data.

Experimental design

It meets the definition of original primary research.

The research question could be better defined - how does it advance the finding of the previous 2017 paper. What is the new research question that the current manuscript aims to answer (that the previous one did not)?

The investigation is building on the previous publication by the group and appears to interpolate dietary assessment onto the previous findings to explain the observations of urinary microbiome variations and IL-8. The technical and ethical standards appear appropriate for the research.

Methods
1. The methods state that and matched healthy control group (HC) were recruited but no data is available in Table 1 on the healthy control group characteristics. It is impossible to judge how well matched the groups were in fact without this data. This data should be added and appropriate statistical group comparisons presented.

2. Why was the first urine of the day sampled (presumably the overnight urine) which likely acts as more of a reservoir for endogenous microbes. How much does this urine sample differ from subsequent urine samples (microbiologically) and does it reflect more closely the faecal / perineal microbiome?

3. Can the authors provide some evidence of accuracy and bias associated with their method of measuring food intake? Have the authors independently validated this tool to accurately quantify habitual fibre and vitamin intake?

Validity of the findings

Many of the conclusions drawn are somewhat circumstantial and not fully supported by the available evidence (presented here and from elsewhere). For example, the discussion around ruminococcus acting as a probiotic through SCFA mediated regulation is plausible in the intestine and has been demonstrated to down-regulate LPS mediated IL-8 secretion. Here however the authors state that "...increase in urinary IL-8 concentration in the bladder may be associated with a higher relative abundance of Ruminococcus". How does this equate to a probiotic effect? Is this not in fact the opposite in that the bacterium promotes IL-8 secretion? The link between urinary microbiota and SCFA is highly speculative in this study as urinary SCFA were not even measured. The primary site of SCFA production is the large intestine and very limited quantities are normally excreted in urine. Without a direct measurement it is not clear how relevant SCFA are in this context and their inclusion to the discussion is rather counter-intuitive given the evidence on SCFA and IL-8. Would a simpler explanation for the ruminococcus data be fecal / perineal microbiome infection of the urinary stream as ruminococcus is associated with fibre (resistant starch) fermentation in the large intestine? What data do the authors have to rule this in or out as an explanation of the findings?

Is the data on the effect of water not more simply explained as a simple dilution effect since urinary IL-8 concentration is measured. It does not appear that the measurement was correct for total urine volume output nor other surrogate marker e.g. creatinine. If the data is expressed relative to a marker of urinary output, do these relationships still stand. This is fairly critical to the interpretation of the findings.

Much is made of the relationships between certain nutrients and their inter-relationship with the urinary microbiome and IL-8. However, the nutrient intake data presented in Table 2 raises some issues as to the quality of the data and therefore the strength of the conclusions. How does the energy and nutrient intake data compare with reference data for this population? Energy intake in particular seems low, particularly for the T2DM group. Is this reporter bias or assessment bias and therefore how does affect the accuracy of the nutrient intake assessment? I don't accept that you can conclude "there may have been some bias regarding the food intake assessments, for example, due to recall issues." without quantifying that bias and you have data that may provide insight into this.

·

Basic reporting

This is a well written paper which shows a very good understanding of the field. Extensive literature searches have been done and there is good use of these to back up the work undertaken. Good evidence of experimental work is evidenced in well presented tables and figures. The paper is well structured and provides clear answers to the hypotheses.


I have no suggestions for improvement to the paper, I feel it meets the standards of the paper.

Experimental design

The research question is well defined and good explanations are given as to the background to the experimental work.

The experimental work has been done to a high standard and ethical approvals were obtained ahead of the research.

Methods are well explained and are clear to understand.

There is no need for improvement as I feel the paper meets the standards required.

Validity of the findings

The results are well presented and provide robust evidence with good statistical analysis.
Discussion of the results are clear and provide good explanations backed up with reference to other publications where needed.
Conclusions are clear and relate to the research question and definitely raise some interesting points that would be interesting to pursue in the future.


No need for improvements as feel the paper meets the standards.

Additional comments

This is a very interesting paper which is well written and shows good experimental design and analysis of the data. It definitely shows some interesting links with diet, bacterial species and IL-8 and expands the understanding of the interactions between the urinary microbiome and the bodies response to UTI.

Reviewer 3 ·

Basic reporting

Needs some major grammatical improvements through the manuscript.

The authors did not provide enough field background. More literature references should be reviewed in the introduction to support the hypotheses.

Experimental design

no comment

Validity of the findings

Line 265, the authors claim that “…help us to recognize the onset of bladder infection in T2DM patients…”. Is it possible to track the occurrence of UTI in these cases in the following period of time?

Line 258, To add the quantitative results of SCFAs is recommended to support the findings in this article. The same problem also exists in the following paragraphs.

Additional comments

In Fig S1 S2 S4, the data showed the high sequences proportion of Lactobacillus and Prevotella. These two genera are the common bacterium in female genital tract. Is it contaminant or is it true in the bladder? What controls were in place for this?

The table S2 was missing.

To explain the co-effect of these genera which were referred in the discussion.

Inappropriate literature citation. The time interval between the references, Tian et al., 2016 (Line273) and Scholes et al., 2000; Beetz, 2003(Line 275), is too long to be compared.

---

## Round 0.2 · Minor Revisions

Thanks for attending to the issues raised. Reviewer 1 has raised a few points which we would like you to address before final acceptance. I don't think these will present you with much difficulty. I hope you will be able to do this and re-submit.

·

Basic reporting

The revised manuscript by Liu et al is much improved and is well structured and clear.

Experimental design

The experimental design builds on previous work by this group and the methods are well described.

Validity of the findings

Thank you for adding the additional data on the HC group. However, there are some perplexing data here. The anthropometric / demographic data for the groups are identical across many of the parameters - that is somewhat surprising but may indicate that the team matched their cohorts extremely well. It should be clear in the methods how these cohorts were recruited and any selection bias - they seem too well matched than is usually possible in such human studies.

I am still perplexed by the energy intake data. The HC had a significantly higher energy intake but not statistically different BMI. Exercise appeared to be identical between the groups which leaves an "energy gap" that needs explaining. Either the T2DM group expended less energy (not captured by the data) or they under-reported their dietary intake which might be a cautionary note on interpretation of the dietary intake data. I don't doubt the validity of their intake data in the study period - just how representative it may be of habitual intake. Would this be expected to impact on the interpretation of the study? Would a change in diet - sometimes seen when participants partake in dietary intake research - impact on the urinary microbiome and inflammatory markers?

Additional comments

No additional comments.

Reviewer 3 ·

Basic reporting

No comment

Experimental design

No comment

Validity of the findings

The new information presented is sufficient to support the findings and the conclusions.

---

## Round 0.3 · accepted · Accept

Thank you for your attention to the outstanding comments; I am delighted to recommend acceptance of the revised paper.